# Asymptomatic gallstones: Cumulative incidence proportion, incidence rate, and risk factors for symptoms development: Systematic review and meta-analysis

Mohammad Alzoubi[1], Ahmad Omar Saleh[1], Farah Al Omari[1], Kinda Shatnawi [1]*, Batool Hyari[1], Ala'a Qashou[1], Khaled Daradka[1], Salam Daradkeh[1], Mohammad Abu Hilal[1,2], Alessandro Parente[3]

1 Department of General Surgery, The University of Jordan, Amman, Jordan, 2 Department of Surgery, University Hospital Southampton NHS Foundation Trust, Southampton, United Kingdom, 3 Institute of Liver Studies, King's College Hospital NHS Foundation Trust, London, United Kingdom

* kindashatnawi3@gmail.com

## Abstract

### Objectives

This review aims to evaluate the cumulative incidence proportion, incidence rate, and risk factors for progression of incidentally diagnosed, asymptomatic gallstones to symptomatic gallstone disease (GSD) and associated complications.

### Design

Systematic Review and Meta-Analysis.

### Data source

Four electronic databases were searched (PubMed, Scopus, Web of Science, and ScienceDirect) with no start date restriction, up to July 2025.

### Eligibility criteria

Inclusion criteria: patients who were diagnosed with gallstones incidentally. Exclusion criteria: known history of GSD, patients who have undergone bariatric surgery or cholecystectomy, recurrence of gallstones, pregnancy, estrogen therapy, pediatric age group, review, case report, case series, editorial, letters, and abstracts.

### Data extraction and synthesis

This review is registered with PROSPERO (CRD42024526889). Primary screening by title and abstract was conducted in Rayyan; full-text screening was performed, and the references of the included studies were manually searched for relevant papers. Data were extracted into an Excel sheet, and the meta-analysis was

**Data availability statement:** All relevant data are within the manuscript and its Supporting information files.

**Funding:** The author(s) received no specific funding for this work.

**Competing interests:** The authors have declared that no competing interests exist.

conducted using RStudio. Single-arm outcomes were summarized in proportion, and comparative outcomes were summarized in Risk Ratio (RR) for categorical outcomes and mean difference for continuous ones. Heterogeneity was evaluated using the I statistic and the Q test.

## Results

Eight cohort studies, reported in 9, with a total of 25,924 participants, were included. The cumulative incidence proportion of symptomatic progression was 0.10, 95% CI: [[0.10; 0.11]] at 5 years, 0.19, 95% CI: [0.14; 0.25] at 10 years, and 0.26, 95% CI: [0.12; 0.40] at 15 years. Alcohol consumption (RR: 1.32, 95% CI: [1.27; 1.38]) and hyperlipidemia (RR: 1.19, 95% CI: [1.07; 1.32]) were identified as risk factors. Chronic liver disease (RR: 0.76, 95% CI: 0.67; 0.87) and male gender (RR: 0.54, 95% CI: 0.33; 0.87) were observed as protective factors.

## Conclusion

This systematic review examines factors influencing symptomatic progression of ASG and guides the identification of high-risk patients who may benefit from prophylactic measures such as cholecystectomy.

---

## 1. Introduction

Gallstone Disease (GSD) is one of the most common and costly digestive diseases worldwide [1], with a prevalence of up to 15% in adults [2,3]. However, 80% of GSD [4] are incidentally detected during imaging without prior recognition of symptoms, making them asymptomatic gallstones (ASG) [4,5].

The prevalence of ASG as an independent subset varies across populations, ranging from 2.5% in Iran [6] to 12.1% in China [7]. It reaches higher rates in exceptional circumstances, such as 53% in post-bariatric surgery patients and 54% in cirrhotic patients [4]. Moreover, the prevalence has been on the rise in recent years, due to the widespread use of ultrasonography [8,9], and the sharp increase in morbid obesity rates [10–12]. This heterogeneity highlights both the lack of and the need for clear and consistent epidemiological reporting.

The natural history of ASG generally follows a benign course, as only about 1–4% of ASG patients develop symptoms each year [3,4,13], with a lifetime risk of 25–33% [8,14]. However, this progression is clinically significant as it marks the transition to symptomatic biliary disease. The transition could range from mild self-limiting biliary colic to more serious conditions, including acute cholecystitis [5,8,14] or, rarely, gallbladder cancer [10], placing a burden on healthcare systems. This broad spectrum of severity matters because it can influence decision-making regarding surveillance, management, and patient counselling.

Despite the relatively high prevalence of ASG, the clinical focus has been diverted to the symptomatic group, leaving the course, possibly influencing risk factors and

outcomes of the ASG natural history, not clearly identified, thereby limiting proactive decision-making. Moreover, the studies addressing this topic are very few, and report widely variable prevalence and progression rates, with inconsistent reporting of risk factors. This lack of clarity represents a significant gap in clinical practice, and we saw the need for a comprehensive review to address all aspects in a single, unified source rather than relying on scattered findings. Such a review could reduce the burden on healthcare systems and highlight research gaps related to this topic, particularly because no prior systematic review has addressed the incidence rate, cumulative incidence, and progression risk factors together.

This study aims to assess the cumulative incidence, the incidence rate of progression of ASG, and the risk factors associated with the symptom development in GSD, directly addressing the identified gaps in current literature, thus offering additional evidence to help physicians in their decision-making when dealing with this increasingly common finding, and to guide management strategies.

## 2. Materials and methods

### 2.1. Search strategy

This review was registered with PROSPERO (CRD42024526889) and followed Preferred Reporting Items for Systematic reviews and Meta-Analyses (PRISMA) guidelines [15]. Utilizing a comprehensive search of four electronic databases (PubMed, Scopus, Web of Science, and Science Direct) from their inception to July 2025, with no date filter applied. The search was done using the following terms: "asymptomatic gallstone"OR, "silent gallstone" OR, "silent cholelithiasis" OR, "asymptomatic cholelithiasis" OR, "non-complicated gallstone" OR, "Incidental gallstone" OR, "non-complicated cholelithiasis" OR, "Incidental cholelithiasis" in English language articles only, as shown in S1 Table.

### 2.2. Study selection and data extraction

Primary screening by title and abstract was managed using Rayyan [16], and duplicates were removed initially. Two independent authors (FAO and KSh) screened titles and abstracts to assess eligibility for inclusion, and conflicts were resolved by consensus among a third independent author (AOS). Afterwards, the following authors conducted full-text screening (FAO, BH, KSh, AOS, AQ). Lastly, the references for the included studies were manually searched for relevant papers (FAO, BH). The eligibility criteria for our systematic review were the following: Inclusion criteria: Population (P): adult patients who were found to have asymptomatic gallstones by imaging done for reasons unrelated to the biliary system; Exposure (I): refers to the development of symptomatic gallstones from previously asymptomatic gallstones, as well as exposure to possible clinical, demographic, and metabolic risk factors; Comparison (C): patients with asymptomatic gallstones who did not develop symptoms; Outcome (O): Progression to symptomatic gallstone disease and related complications, measured as cumulative incidence proportion, incidence rate, and associated risk factors.. Exclusion criteria: known history of GSD, patients who have undergone bariatric surgery or cholecystectomy, recurrence of gallstones, pregnancy, estrogen therapy, pediatric age group (below the age of 16 years), review, case report, case series, editorial, letters, and conference abstracts. Bariatric surgery patients were excluded due to extensive prior research and literature coverage, as our review focuses on the general population.

The country, study design, total cohort size, sample sizes of asymptomatic and symptomatic groups, age, gender, length and duration of follow-up, diagnostic method, setting (hospital- or community-based), and cumulative incidence proportion were extracted by two independent authors from each study into a structured Excel sheet.

### 2.3. Study outcomes

The primary outcome of interest was the cumulative incidence proportion of symptomatic gallstones from known ASG at specific time point interval of 5, 10, and 15 years, the incidence rate, and the factors that influence conversion from

asymptomatic into symptomatic disease, which included: age, gender, smoking, alcohol consumption, hyperlipidemia, diabetes mellitus, chronic liver disease and gallstone number ≤ 2. The secondary outcome was to determine the risk of symptomatic GSD and the proportion of this complication. Complication events were classified as either complicated or non-complicated. Biliary pain was considered non-complicated, whereas the complicated events included: common bile duct (CBD) stones, acute cholecystitis, gallstone pancreatitis, adenocarcinoma of the gallbladder, and obstructive jaundice.

## 2.4. Quality assessment

A subjective assessment of methodological quality for observational studies was evaluated by two authors using the Newcastle-Ottawa Scale (NOS). This tool is used for assessing the quality of nonrandomized studies included in systematic reviews and/or meta-analyses. Each study is estimated using a 'star system' across eight items categorized into three domains: 1) selection of the study groups (0–4 stars), where a score of 4 was considered high, three or below was considered moderate, and zero was considered low. 2) Comparability of the study groups (0–2 stars), where two stars were considered high, 1 star was considered moderate, and zero stars was considered low. 3) Ascertainment of the outcome of interest (0–3stars), where three stars were considered high, 2 or 1 stars were considered moderate, and zero was considered low. While the total score, which is the most important, was calculated by the summation of the previous three scores, and was classified as poor, moderate, and high quality, depending on the following classification: a score of ≤ 3 was considered inadequate, 4–6 was considered moderate, and 7–9 was deemed high quality [17].

## 2.5. Statistical analysis

Meta-analyses were conducted in RStudio [18] when studies reported sufficient data, and systematic reviews were undertaken when insufficient data were available for meta-analysis. A single-arm meta-analysis was performed to estimate complication proportions and the cumulative incidence proportion for progression to symptomatic gallstone disease, using a generalized linear mixed-effects model. Cumulative incidence was also conducted using the meta function [19]. A comparative meta-analysis was performed using the Mantel-Haenszel method and expressed as a Risk Ratio (RR) to assess the risk factors for symptomatic gallstone disease and the risk of gallstone complications [20,21]. A leave-one-out sensitivity analysis was performed to evaluate the robustness of the results by exploring the contribution of each study to the pooled effect and heterogeneity. Publication bias was assessed using funnel plots, the Luis Furuya-Kanamori (LFK) index, and a DOI plot.

Statistical heterogeneity was evaluated using the I² statistic and the Cochrane Q test. I² values were interpreted as 0–40% (not essential), 30–60% (moderate heterogeneity), 50–90% (substantial heterogeneity), and 75–100% (considerable heterogeneity) [22,23]. The Cochrane Q test p-value <0.05 was considered statistically significant for heterogeneity. A random-effects model was applied to all analyses, regardless of observed heterogeneity, to account for clinical and methodological variability across studies [24]. Statistical significance was assessed using P values <0.05.

## 2.6. Ethics statement

This study is a systematic review and meta-analysis of previously published studies. Therefore, ethical approval and informed consent were not required.

## 3. Results

### 3.1. Study selection

The initial systematic literature search retrieved 10579 studies, of which 2130 were excluded from this meta-analysis due to duplication. Subsequently, 8356 studies were excluded after title and abstract screening as they did not meet the

research outcomes. Full-text screening included 93 studies, of which 84 were excluded due to being abstract-only, having no reported results, being reviews, or reporting incorrect outcomes. Finally, eight Retrospective and prospective Cohort studies that were reported in 9 [13,25–32]- as the Danish study was reported in two studies [26,29] were included in qualitative and quantitative analysis [13,25–32]. As shown in Fig 1 the flowchart below shows the study selection process.

For publications written in languages other than English, the primary screening process identified 402 non-English papers with English abstracts. All were excluded for being outside the study scope, involving pediatric populations, discussing procedures like cholecystectomy, or classified as reviews or case reports. Additionally, two studies written in German and Japanese were excluded after an extensive search, as they were inaccessible and lacked abstracts.

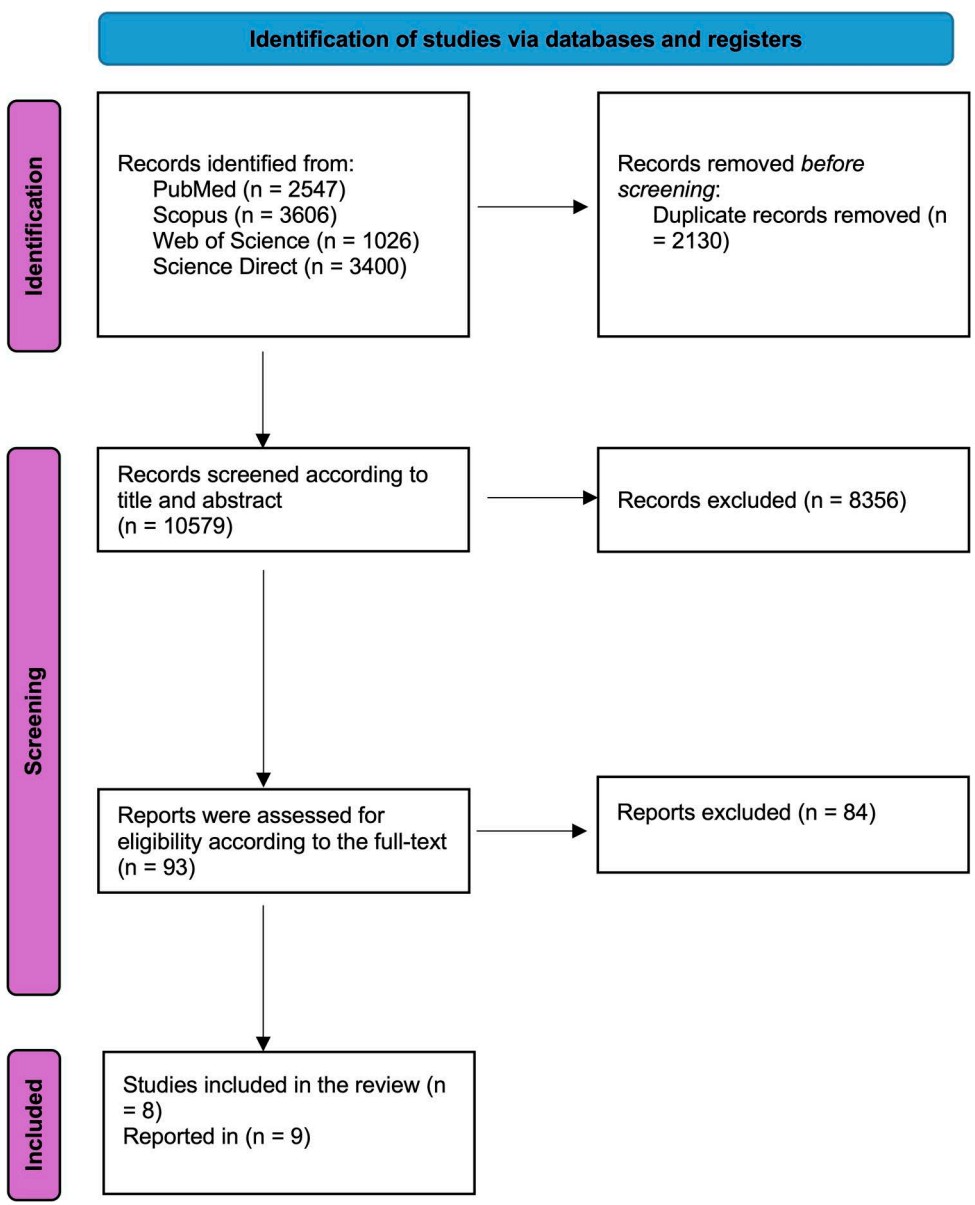

**Fig 1. PRISMA 2020 Flow Diagram of Study Selection.**

### 3.2. Summary of Included Study and population characteristics

The summary of the studies included is presented in Table 1—data from 24828 (range 109–22257) of ASG patients. The mean age of asymptomatic patients was 55.53 (range 20–87) years. The percentage of females gender ranged from 10.55% to 71.5%. The cumulative incidence percentage at 5 years ranged from 6% to 28.45%; at 10 years, from 11% to 32.61%; at 15 years, from 16% to 34%; at 20 years, from 18% to 41%; and at 25 years, from 20%. The summary of included studies table also contains the diagnostic methods used in each study, setting, and follow-up period.

### 3.3. Quality assessment of included study

S1 Fig represents the Newcastle–Ottawa scale of the included studies. After the quality assessment, four studies [25–28] had moderate quality and five [13,29–32] had high quality in the selection domain. Two studies [27,28] and seven studies [13,25,26,29–32] showed low and high quality of comparability domain, respectively. Two studies [25,27] and seven studies [13,26,28–32] showed moderate and high quality of outcome domain, respectively.

The overall quality rating of the included studies: three [25,27,28] showed moderate quality, and six [13,26,29–32] showed high quality.

### 3.4. Outcomes

**3.4.1. Cumulative incidence and incidence rate (IR).** Incidence rate was 0.02 per person-year (95% CI: 0.01; 0.05, $I^2 = 93.6\%$), with high heterogeneity, S2 Fig. Leave-one-out sensitivity analysis supported robustness of pooled rate after excluding each study, and heterogeneity decreased into $I^2 = 61.7\%$ after omitting Shabanzadeh, DM (2016) S3A Fig. Visual inspection of the funnel plot indicated an asymmetrical distribution of studies, with statistical evidence of publication bias (LFK = 4.56), S4A Fig.

Cumulative incidence was assessed at 5, 10, and 15 years Fig 2. At 5 years of follow-up, the cumulative incidence proportion was 0.10 (95% Confidence Interval (CI) [0.10; 0.11], $I^2 = 0\%$); results for this outcome had low heterogeneity. Leave-one-out sensitivity analysis confirmed that the pooled proportion and heterogeneity remained consistent after excluding each study S3B Fig. Visual inspection of the funnel plot indicated an asymmetrical distribution of studies, with statistical evidence of publication bias (LFK = −6.44; S4B Fig).

At 10 years of follow-up, the cumulative incidence proportion was 0.19 (95% CI [0.14; 0.25]), with moderate heterogeneity ($I^2 = 61.8\%$). At 15 years of follow-up, the cumulative incidence proportion was 0.26 (95% CI [0.12; 0.40]), with high heterogeneity ($I^2 = 92.2\%$). Leave-one-out sensitivity analysis was not feasible due to the limited number of studies.

**3.4.2. Risk factors.** We assessed nine risk factors that may influence the development of symptomatic GSD from ASG. Herein, all risk factors were evaluated by two or more studies, as shown in Fig 3, which reports RR, CI, P-value, and $I^2$ for each risk factor.

Two factors, alcohol consumption and hyperlipidemia, were found to be statistically significant risk factors for the development of symptoms from previous ASG. The results for alcohol consumption were as follows: RR: 1.32, 95% 1.27; 1.38, and $I^2 = 0\%$. For hyperlipidemia, they were RR: 1.19, 95% 1.07; 1.32, and $I^2 = 0\%$. Both factors had low heterogeneity, and leave-one-out sensitivity analysis was not feasible due to the limited number of studies.

On the other hand, another two factors, Chronic liver disease (CLD) and male gender, were found to be statistically significant protective factors against the development of symptoms from previous ASG. The results for male gender were as follows: RR: 0.54, 95% 0.33; 0.87, and $I^2 = 84.4\%$. For chronic liver disease, the results were as follows: RR: 0.76, 95% CI: 0.67–0.87, and $I^2 = 0\%$ with low heterogeneity.

The remaining five factors, were found to have no statistical significance of being a risk factor for symptoms development from previous ASG, which include: female gender RR: 0.93, 95% CI: 0.61; 1.40; $I^2 = 93.2\%$, smoking RR: 1.05, 95% CI: 0.94; 1.38; $I^2 = 40.5\%$, diabetes mellitus (DM) RR: 0.98, 95% CI: 0.91; 1.05; $I^2 = 0.7\%$, gallstones number less than or equal to 2 RR: 8.13, 95% CI: 0.16; 415.51; $I^2 = 97.1\%$, and age MD: −0.49, 95% CI:-15.51; 14.53; $I^2 = 98.2\%$. The results of

Table 1. Summary of Included Studies.

| Study Id | Country | Study Design | Total Cohort Size | Sample Size of Asympto-Matic | Sample Size of Symptom-Atic | Age, mean (range) | Gender (female), % | Median (Range) of Follow up | Diagnostic Method | Setting (Hospital or Community) | Follow-up Duration | Cumulative Incidence proportion at year |
|---|---|---|---|---|---|---|---|---|---|---|---|---|
| **Gracie et al. 1982** [28] | The United States | prospective cohort | 1233 | 123 | 0 | 54 (29-87) | 71.50% | 1959 - 1980 | cholecystography | hospital | 21 years | At 5 years: 10% At 10 years: 15% At 15 years: 18% At 20 years: 18% |
| **Friedman et al. 1989** [27] | The United States | prospective cohort | 467 | 132 | 298 | 53 | 55% | 1967 - 1994 | cholecystography, palpation during non-biliary abdominal surgery, and self-administered questionnaire. | hospital | 25 years | At 5 years: 18% At 10 years: 30% At 15 years: 34% At 20 years: 41% |
| **Halldestam et al. 2004** [30] | Sweden | cohort study | 120 | 120 | 0 | (35–85) | 60.833 | 7.25 (0.25 - 12.17) | ultrasonographic screening for gallbladder stones | **community – in a mixed urban–rural municipality of Linkoping, Sweden,** | 12 years | At 5 years: 7.6% |
| **Thwayeb et al. 2004** [32] | spain | prospective cohort study | 138 | 109 | 0 | 55 (21–87) | – | 1982-2003 | Abdominal ultrasound and medical records | Community (participants were selected randomly from canary Island) | 18 years | At 5 years: 28.45% At 10 years: 32.61% |
| **Sood et al. 2015** [25] | Malysia | retrospective cohort study | 213 | 213 | 0 | 58 | 58,10% | mean (range) 4.02(0.077–11.77) | interview, abdominal ultrasonography | government hospital and two private clinics | 9 years (2001–2010) | At 5 years: 13.81% At 10 years: 13.81% |
| **Shabanzadeh et al. 2016** [26] | Denmark (Copenhagen) | **cohort study** | 664 | 664 | 0 | 60(50-65) | 50.90% | 17.4 (0.1–29.1) | **. Participants appeared after 12 hours of fasting and underwent a general health examination, including a questionnaire about lifestyle and medical history, and had an abdominal ultrasound examination to assess gallstone status.** | **General population of urban Copenhagen** | 29 years | At 5 years: 6% At 10 years: 11% At 15years: 16% At 20 years: 18% At 25 years: 20% |

*(Continued)*

Table 1. (Continued)

| Study Id | Country | Study Design | Total Cohort Size | Sample Size of Asymptomatic | Sample Size of Symptomatic | Age, mean (range) | Gender (female), % | Median (Range) of Follow up | Diagnostic Method | Setting (Hospital or Community) | Follow-up Duration | Cumulative Incidence proportion at year |
|---|---|---|---|---|---|---|---|---|---|---|---|---|
| Shabanzadeh et al. 2017 [29] | copenhagen, denmark | cohort study | 595 | 595 | 0 | (30-70) | 51.30% | median 17.5 years | ultrasound examinations | community | examined between 1982–1994 and followed up until December 31, 2011. | |
| Morris-Stiff et all. 2023 [13] | The United States | retrospective cohort | 22257 | 22257 | 0 | 61 | 48.1% | median 4.6 (1.8–7.9) | programmed AI chosen records of abdominal ultrasounds or CT scan of gallstones from 1996 to 2016 | hospitals and medical centers | 10 years (1996–2016) | At 5 years: 10.1% At 10 years: 21.5% At 15 years: 32.61% |
| Sakai et al. 2024 [31] | China | Longitudinal observational study | 237 | 237 | 0 | 47.675 (20-78) | 10.55% | 50-4111 days, 10.7 years on average | abdominal ultrasonography performed during screening | Hospital (A bedless clinic) | (12 years) Between March 2010 and October 2022 | |

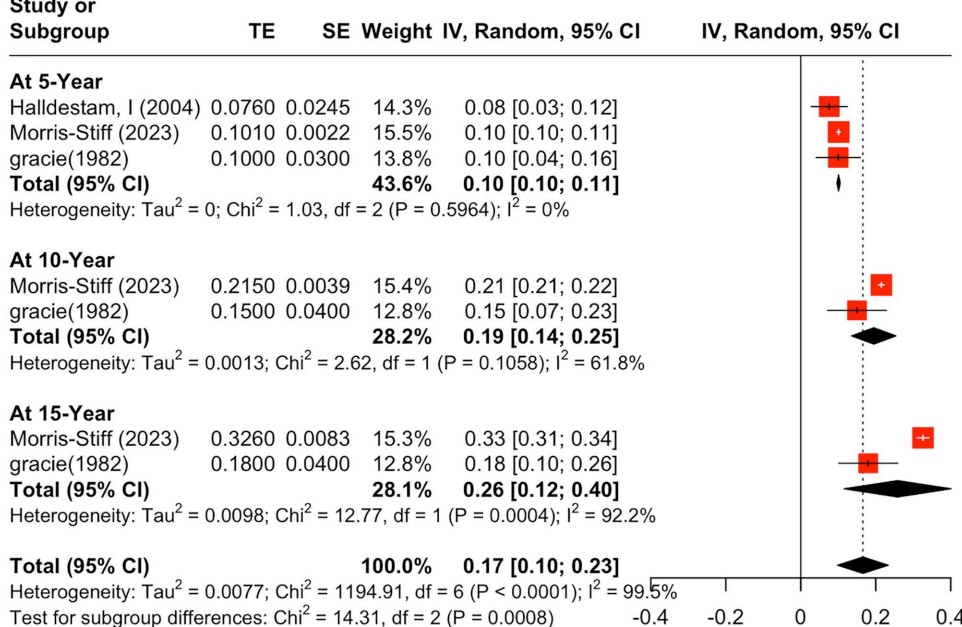

**Fig 2. Forest Plot of Cumulative Incidence Proportion at Different Time Intervals.**

the analysis for each risk factor are shown in Figs 3 and S5. The heterogeneity among these five factors was variable. It was low for DM, moderate for smoking, and high for female gender, number of gallstones, and age.

Assessment of publication bias and leave-one-out sensitivity analysis are presented in the Supporting Information (S1 Appendix).

**3.4.3. Complications.** Complicated events shown in S6 Fig showed no statistically significant difference in presenting patients with complicated gallstone disease, according to these statistics; RR: 1.57, 95% 0.64; 3.89, $I^2$ = 86.4%. The heterogeneity of this outcome was considered high.

Complications of conversion from ASG are shown in S7 Fig. The overall proportion of patients presenting with biliary pain was 0.60 (95% CI [0.26; 0.86], $I^2$ = 87.2%). The heterogeneity of this outcome is considered high. The overall proportion of patients presenting with common bile duct stones was assessed by six studies [13,26,27,29–31]: 0.19 (95% CI [0.17; 0.20], $I^2$ = 44%). The heterogeneity of this outcome is considered moderate. The overall proportion of patients with Acute cholecystitis was 0.19 (95% CI [0.12; 0.29], $I^2$ = 74.1%). The heterogeneity of this outcome is considered moderate. The overall proportion of patients with gallstone pancreatitis was 0.07 (95% CI [0.03; 0.14], $I^2$ = 72.5%); the heterogeneity of this outcome is considered moderate. Overall, the proportion of patients presenting with gallbladder adenocarcinoma was 0.03 (95% CI [0.02; 0.04], $I^2$ = 0%). Results for this outcome can be considered reliable, given the low heterogeneity. Overall, the proportion of patients presenting with obstructive jaundice was 0.06 (95% CI [0.03; 0.15], $I^2$ = 0%). The heterogeneity of this outcome is considered high. Assessment of publication bias and leave-one-out sensitivity analysis are presented in the Supporting Information (S1 Appendix).

## 4. Discussion

This study has several strengths; it is the first systematic review and meta-analysis to evaluate the incidence and risk factors of symptomatic GSD development in patients previously diagnosed with ASG. This study combines the results of all large studies on this topic published between 1950 and 2025.

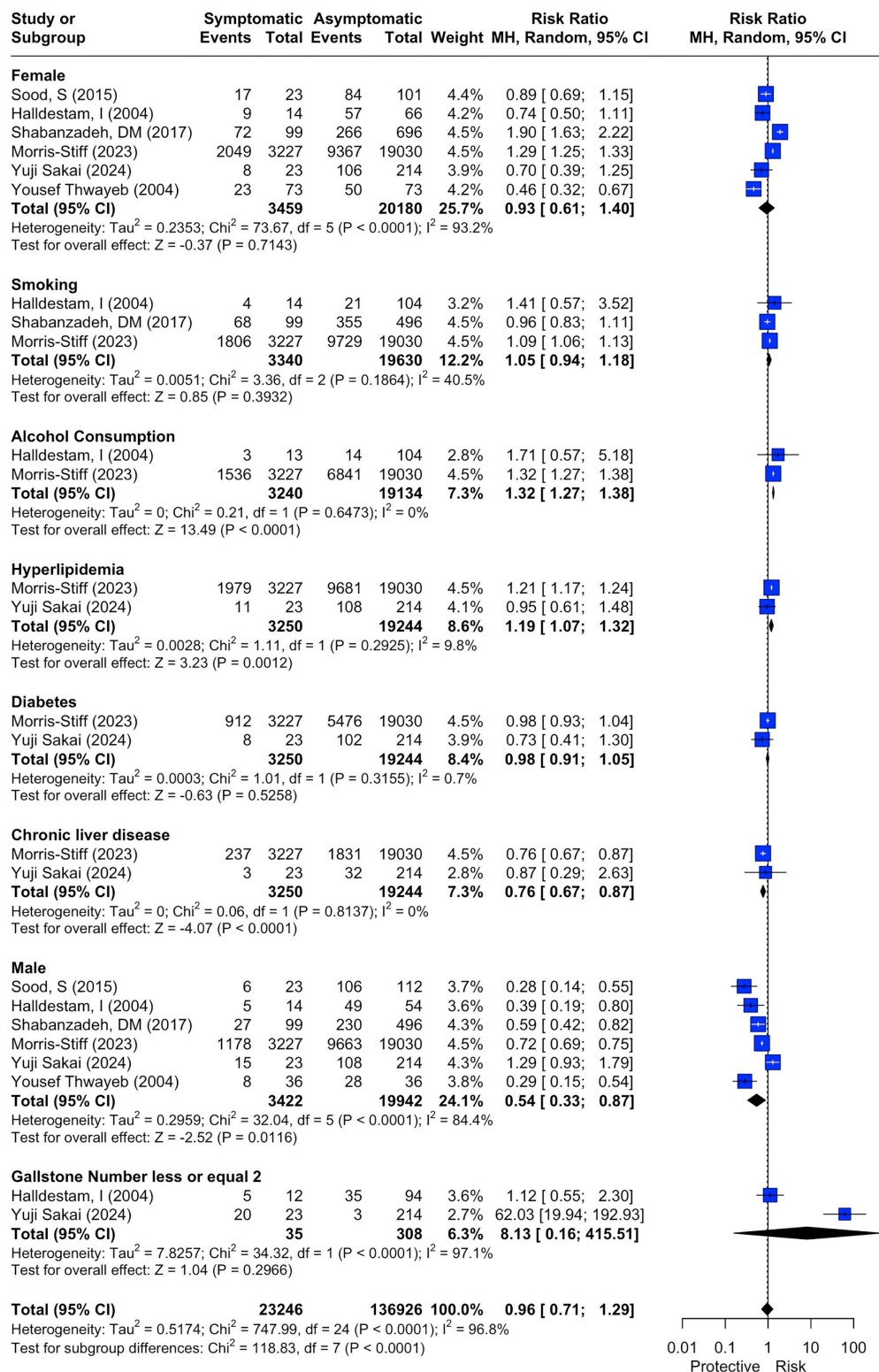

**Fig 3. Forest Plot of Risk Factors for Symptoms Development from Previous ASG.**

The comprehensive literature search included manual searches of the reference lists of the included studies, a precise manual data extraction process, and sensitivity analyses. All of which strengthened the results and enhanced their integration into clinical practice. This study provides valuable insights into who is likely to develop symptoms over time from ASG patients, and who would benefit from a prophylactic cholecystectomy or who would require a customized follow-up plan.

The first finding was an incidence rate of 0.02, meaning that among 100 patients with asymptomatic gallstone disease, two will develop symptoms each year.

This finding is consistent with previously reported annual risk of symptom development, which ranges from 1% to 4% [3,4,13]. This finding was further supported by the leave-one-out sensitivity analysis, which confirmed robustness of the pooled rate, while showing that the reason behind the high heterogeneity was mainly attributed to a single study (Shabanzadeh et al. 2016 [26]); once this study was excluded, the heterogeneity dropped significantly, while the pooled rate remained stable. This is likely due to the markedly larger sample size and longer follow-up duration of this study, which may have led to an overestimation effect compared to the other two studies [25,28]. This result suggests that most patients with ASG remain symptom-free over time, which supports the decision for a conservative management approach for low-risk patients.

The cumulative incidence proportion increased progressively over longer follow-up durations; 5, 10, 15 years, and the reason behind choosing these follow up periods is because only three studies (Halberstam I, 2004; Morris-Stiff, 2023; Gracie, 1982) [13,28,30] reported the effect size along with the corresponding standard error (SE), while the other studies provided only the effect size. However, the increase in cumulative incidence is explained as follows: More participants develop symptoms over time. An increase in heterogeneity was also noted, due to differences in the studies' populations: one study was community-based [30], whereas the other two were hospital-based [13,28]. A leave-one-out sensitivity analysis at a 5-year interval confirmed that the pooled proportion and its low heterogeneity remained stable after excluding each study, supporting this result at this time point. For 10 and 15 years, the cumulative incidence proportion was assessed by only two studies, with moderate heterogeneity at 10 years and high heterogeneity at 15 years. The robustness of these results couldn't be tested because the leave-one-out sensitivity analysis requires at least three studies per factor.

As for the assessed risk factors, the most significant predictors of symptom development included alcohol consumption and hyperlipidemia. Conversely, chronic liver disease and male gender were observed as protective factors. Other factors like female gender [13,25,29–32], smoking [13,29,30], diabetes mellitus (DM) [13,31], gallstones number less than or equal to 2 [30,31], and age [13,29,32] were not found to have any statistical significance as risk factors (Figs 3 and S6).

Moderate alcohol consumption is known to have a protective effect on the development of GSD [33,34], as it decreases bile cholesterol saturation [35–37] and elevates hepatic bile salt production and excretion [38,39]. Controversially, a negative effect of alcohol consumption on GSD was reported by different studies due to its link with hepatobiliary diseases in general, namely cholangitis and alcoholic liver disease [40], and to the effect of alcohol in slowing down gallbladder emptying [41]. In general, the symptom development in patients with previous ASG seems different from that in patients with de novo ASG; the two states are difficult to compare. Consequently, the potential protective effect of alcohol consumption on asymptomatic gallstone carriers has not been mentioned before in existing literature, even though another study suggests that it does not alter the clinical course [42].

This highlights a gap in understanding that warrants further investigation. The present analysis has interestingly identified alcohol consumption as a statistically significant risk factor for the development of symptoms from previous ASGs, with low heterogeneity between the two studies that the analysis relied on [13,30]. Although limited to these studies only, the consistency of the findings provided confidence in this observation.

The second significant symptoms predictor found, was hyperlipidemia, this association is consistent with its established negative impact on GSD in general, as serum lipid levels are associated with GSD [43] leading to high levels of secretion of cholesterol in bile, with the subsequent supersaturation of bile contributing to the formation of bile mud, adding that

increased triglycerides level reduces the sensitivity to cholecystokinin (CCK) thus reducing the gallbladder movement [44,45].

Chronic liver disease (CLD) has been associated with increased prevalence of ASG, due to the increased exposure to diagnostic imaging methods in the process of diagnosing CLD [46]. However, CLD [46] and cirrhosis [1] have not been found to predict the clinical course of gallstone disease in a cohort with screen-detected gallstone disease, and here we also noted the need for further studies to address this.

Men are, in general, prone to have more severe GSD [47], but interestingly, this study found that being male was associated with a lower risk of developing symptoms than being female. While female sex is considered a frank risk factor of developing gallstone disease in general [47], female sex was found to be in the no effect group in the present study. However, It is important to emphasize on that this study is investigating the development of symptoms in an individual with existent asymptomatic gallstones, not the etiology of gallstones formation for which female gender is an established risk factor, adding to that the fact that the mean age of females in all the studies that were involved into the analysis was 52 years, an age at which estrogen that is the main contributors to the etiology of cholesterol gallstones in females [48] is greatly reduced, this observation could also affect the symptomatic presentation by affecting the etiological mechanisms behind gallstones.

Smoking, DM, and gallstones number ≤2 are expected to be associated with an increased likelihood of symptom development, but they were not found statistically significant as risk factors in symptom development from ASG. This discordance could be due to hidden mechanisms yet to be exposed regarding the unpredictable course of ASG, or to the small number of studies exploring them as risk factors, highlighting the need for more studies covering these topics in depth.

A single study [13] has reported Hispanic ethnicity as a protective factor. In contrast, other factors, including inflammatory bowel disease (IBD), Chronic hemolytic disease, Gallbladder Sludge, Multiple gallstones, polyps, nonspecific thickening of the gallbladder wall, and gallstones larger than 9 mm, were all reported as statistically significant risk factors for developing the symptomatic disease.

Sensitivity analysis was conducted for each factor using the leave-one-out model. Significant results from this analytic tool were observed for 4 of the nine factors: male gender, female gender, age, and smoking. The remaining five factors could not be assessed because only two or fewer studies addressed them.

For the male gender, the analysis confirmed an association with a protective effect. In contrast, female gender showed no significant impact on symptom development, supporting the robustness of the pooled results despite persistently high heterogeneity for both factors. This high heterogeneity could not be attributed to specific studies but was more likely due to differences in study demographics, geographic distribution, and follow-up durations. There is no scientific evidence proven on why male gender might be a protective factor against developing symptoms in ASG patients yet, so the interpretation of this finding is attributed to statistical bases, as some studies had unequal gender representation in their sample population. Notably, the study by Yuji Sakai (2024) [31] is a clear example, which included only 10.55% females of its population, and still accounted for the most significant drop in heterogeneity – although it remained high overall – when male gender was analyzed, its effect is likely attributable to the study's disproportionate weighting relative to the male population percentages that were reported in the other included studies.

Age showed an interesting shift in effect, with excluding Thwayeb et al. 2004 [32] associated with an increased risk of symptomatic gallstones in patients younger than fifty-five. This study reported a mean age of 60 years among symptomatic cases, whereas the other two studies [13,29] reported lower means of 50 and 54, respectively. The removal of this study significantly reduced heterogeneity, though it remained high.

Another surprising shift was also noted with smoking, when Shabanzadeh et al. 2017 [29] was excluded, smoking emerged as a significant risk factor for symptom development, with heterogeneity dropping to $I^2 = 0$. This finding indicates that Shabanzadeh et al. and Morris-Stiff et al. 2023 [13] were the main contributors to heterogeneity. However, the shift in effect could not be explained clearly. It may be related to the lack of a clear definition of "smoker" or the absence of a

numerical definition in terms of Pack-Years. However, the danish study Shabanzadeh et al. 2017 is a well-known reputable study, mainly due to its large population, long follow up duration, and most importantly the blinding technique which makes its quality undeniably higher than the other studies. In our opinion, this ambiguity, along with the high heterogeneity driven by these two studies [13,29], strongly emphasizes the need for further studies to address the impact of smoking on the natural history of ASG.

Publication bias was evaluated using a funnel plot for all outcomes reported in at least three studies. Visual inspection revealed asymmetry across all plots, suggesting publication bias. This finding was expected, given the limited number of studies addressing this topic, which also limits the reliability of interpreting such asymmetry.

This study highlights the unpredictable nature of asymptomatic gallstone progression and the variability of outcomes under different circumstances, for which an additional outcome was assessed: the nature of presentation upon symptom development, divided into either complicated or uncomplicated events. Complicated events compromise all different symptomatic presentations other than simple biliary pain, which was considered the only uncomplicated symptom. The proportion of each symptom type that developed into symptomatic disease was assessed, along with the risk of presenting first with a complicated event. This risk of first presenting with a complicated event was not statistically significant. This finding could be reassuring, suggesting a low likelihood of serious outcomes with watchful waiting.

In most cases, symptom development presents as simple biliary pain, typically managed with an elective cholecystectomy. The Leave-one-out sensitivity analysis further confirmed the insignificance of this outcome, although heterogeneity remains high even after sequentially excluding studies. The differences in geography and study populations can explain this persistent heterogeneity. Geographic variation raises the genetic influence on the natural history of ASG. At the same time, hospital-based populations may carry additional medical exposures, either protective or risk-enhancing, that are less relevant in community-based cohorts.

It is evident that the articles included in this study have different study designs as well as differences in the population selection techniques, which can potentially affect comparability. The Swedish (Halldestam et al) [30], Danish (Shabanzadeh et al) [29], and the Spanish (Thwayeb et al) [32] are population-based screening studies using the most robust assessment of truly asymptomatic gallstones carriers, as they used ultrasonography in randomly selected individuals regardless of symptoms. In contrast, the rest of the articles where hospital-based studies, this issue was addressed by careful selection of the non-standardized studies. Detecting words, expressions, and phrases that clearly indicate the initial asymptomatic status of the patients before entering the study like "Asymptomatic patients", and "Incidentally discovered gallstones", Moris et al. 2023 for example stated that the study included more than 22,000 patients with asymptomatic gallstones incidentally discovered on imaging [13]. This is assuring that no ambiguous study was included in the analysis.

We furthermore narrowed the inclusion criteria to avoid any potential sources of bias by eliminating populations that have special considerations, such as the rapid weight loss associated with post-bariatric surgery patients and hemolytic diseases that may affect pediatric age group. As it was noted during the literature review that most of the available studies address these specific populations individually; if included, they would result in noticeable selection bias.

Despite the novel concept of this meta-analysis and its strong methodology, certain limitations can't be ignored: First, noticeable heterogeneity in some parts of the results, for which subgroup analysis could not be performed due to insufficient data from the included studies. The limited number of eligible studies, along with the small sample sizes in most included studies, constrained the evaluation of individual risk factors. Heterogeneity is also explained by timely influenced evolution and accuracy of the diagnostic methods used between biliary ultrasound in recent studies and cholecytogram in the 1940s may significantly impact the interpretation of results, differences in data reporting methods, variations in local healthcare systems as geographical differences also limit the ability to perform large and robust studies as evident with the Danish study Shabanzadeh et I,2017 [29] being of higher quality due to its favorable research nationwide database, and genetic predispositions across the included studies. Another important limitation was the variability among the included studies in many aspects of the analyses, including follow-up periods, time intervals for cumulative incidence, and

the types of factors assessed. This was most evident in the cumulative incidence analysis, in which most of the studies reported similar follow-up periods. Still, the absence of Standard Error reporting limited the ability to pool this outcome.

While subgroup analyses could address differences in study populations, this approach was not feasible in the present study because each outcome was reported in at most three studies, and inconsistent reporting prevented the identification of sufficiently comparable factors.

Finally, all the included studies were observational, with no randomized controlled trials identified during the literature review. As such, causal relationships between risk factors and symptomatic gallstone disease cannot be established, since observational studies are susceptible to confounding and selection bias. Therefore, the identified associations should be interpreted as correlations rather than causal relations.

## 5. Conclusion

This systematic review and meta-analysis provide a comprehensive synthesis of the long-term risk of symptom development and complications in patients with previous symptomatic gallstones. The pooled incidence rate of symptom development was low at 0.02, while cumulative incidence increased progressively over time, reaching 10% at 5 years, 19% at 10 years, and 26% at 15 years of follow-up. These findings indicate that although most individuals with asymptomatic gallstones remain symptom-free over time, the risk of developing symptoms increases with longer follow-up.

Alcohol consumption and hyperlipidemia were identified as statistically significant factors associated with an increased risk of symptom development. In contrast, male gender and chronic liver disease were associated with a lower observed risk. However, these findings reflect associations derived from observational data and indicate correlation rather than causality. Other evaluated factors, including diabetes mellitus, smoking, age, female gender, and gallstone number, were not significantly associated with symptom development, although marked heterogeneity was observed in several analyses.

The occurrence of complicated gallstone disease did not differ significantly, and the overall complication rates were low. Biliary colic was the most frequently reported complication, followed by acute cholecystitis and common bile duct stones. In contrast, severe complications such as gallstone pancreatitis, obstructive jaundice, and gallbladder adenocarcinoma were rare.

These findings support a conservative management approach for most patients with asymptomatic gallstones, with particular attention to those who may have a higher risk of symptom development. The findings are limited by heterogeneity across studies and the observational nature of the available evidence, which restricts causal inference. Future prospective studies with standardized definitions and longer follow-up, as well as clinical trials, are needed to refine risk stratification, address gaps in the literature, and guide individualized clinical decision-making.

## Supporting information

**S1 Appendix. Details of the publication bias assessment and leave-one-out sensitivity analysis.**
(DOCX)

**S1 Checklist. PRISMA checklist.**
(DOCX)

**S1 Table. Search strategy and keywords used.**
(DOCX)

**S1 Fig. Newcastle–Ottawa Scale quality assessment of the included studies.**
(TIFF)

**S2 Fig. Forest plot of incidence rate of symptoms development from previous ASG.**
(TIFF)

**S3 Fig. Leave-one-out sensitivity analysis; A) Incidence rate, B) 5-Year cumulative incidence, C) Male gender.**
(TIFF)

**S4 Fig. Funnel plot and DOI plot for; A) Incidence rate, B) 5-Year cumulative incidence, C) Male gender.**
(TIFF)

**S5 Fig. Forest plot of age analysis as a risk factor.**
(TIFF)

**S6 Fig. Forest plot of risk of ASG to present with complicated events vs. non-complicated events (Biliary pain).**
(TIFF)

**S7 Fig. Forest plot of types of events developed from previously ASG.**
(TIFF)

**S8 Fig. Leave-one-out sensitivity analysis: A) Female gender, B) Age, C), Smoking.**
(TIFF)

**S9 Fig. Leave-one-out sensitivity analysis; A) Risk of presenting first with a complicated event, B) Adenocarcinoma of the gallbladder, C) Obstructive jaundice.**
(TIFF)

**S10 Fig. Leave-one-out sensitivity analysis; A) Common bile duct stones, B) Biliary pain, C) Acute cholecystitis, D) Gallstone pancreatitis.**
(TIFF)

**S11 Fig. Funnel plot and DOI plot for; A) Female gender, B) Age, C) Smoking.**
(TIFF)

**S12 Fig. Funnel plot and DOI plot for; Risk of presenting with a complicated event.**
(TIFF)

**S13 Fig. Funnel plot and DOI plot for; A) Biliary pain, b) Common bile duct stones, C) Adenocarcinoma of the gallbladder.**
(TIFF)

**S14 Fig. Funnel plot and DOI plot for; A) Obstructive jaundice, b) Acute cholecystitis, C) Gallstone pancreatitis.**
(TIFF)

## Acknowledgments

The authors acknowledge Dr. Mira Essam Jabri (OBGYN, PGY-2, Jordan University Hospital) for her contribution to the initial phase of the screening process.

## Author contributions

**Conceptualization:** Kinda Shatnawi.

**Data curation:** Ahmad Omar Saleh.

**Formal analysis:** Ahmad Omar Saleh.

**Investigation:** Ahmad Omar Saleh, Farah Al Omari, Kinda Shatnawi, Batool Hyari, Ala'a Qashou.

**Methodology:** Ahmad Omar Saleh, Farah Al Omari, Kinda Shatnawi, Batool Hyari, Ala'a Qashou.

**Project administration:** Mohammad Alzoubi.

**Resources:** Mohammad Alzoubi, Ahmad Omar Saleh, Farah Al Omari, Kinda Shatnawi, Batool Hyari, Ala'a Qashou, Khaled Daradka, Salam Daradkeh, Mohammad Abu Hilal, Alessandro Parente.

**Software:** Mohammad Alzoubi, Ahmad Omar Saleh, Farah Al Omari, Kinda Shatnawi, Batool Hyari, Ala'a Qashou.

**Supervision:** Mohammad Alzoubi, Khaled Daradka, Salam Daradkeh, Mohammad Abu Hilal, Alessandro Parente.

**Validation:** Mohammad Alzoubi, Khaled Daradka, Salam Daradkeh, Mohammad Abu Hilal, Alessandro Parente.

**Visualization:** Mohammad Alzoubi, Khaled Daradka, Salam Daradkeh, Mohammad Abu Hilal, Alessandro Parente.

**Writing – original draft:** Mohammad Alzoubi, Ahmad Omar Saleh, Farah Al Omari, Kinda Shatnawi, Batool Hyari, Ala'a Qashou.

**Writing – review & editing:** Mohammad Alzoubi, Ahmad Omar Saleh, Farah Al Omari, Kinda Shatnawi, Batool Hyari, Ala'a Qashou, Khaled Daradka, Salam Daradkeh, Mohammad Abu Hilal, Alessandro Parente.

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
