## [Decision Letter · Decision Letter 0]

17 Nov 2025

Asymptomatic Gallstones: Cumulative Incidence Proportion, Incidence Rate, and Risk Factors for Symptoms Development: Systematic Review and Meta-Analysis

PLOS ONE

Dear Dr. Shatnawi,

Thank you for submitting your manuscript to PLOS ONE. After careful consideration, we feel that it has merit but does not fully meet PLOS ONE’s publication criteria as it currently stands. Therefore, we invite you to submit a revised version of the manuscript that addresses the points raised during the review process.

We look forward to receiving your revised manuscript.

Kind regards,

Ozlem Boybeyi

Academic Editor

PLOS ONE

Journal Requirements:

2. Thank you for the updates made to your manuscript regarding language restriction, including the description of the screening carried out on non-English language articles. We kindly request that you remove the statements from lines 162-165, as these are generalisations which are not supported by the references in the context of your research.

Additional Editor Comments:

This review gives the analysis of risk factors that may influence the progression of asymptomatic gallstone into symptomatic gallstone disease by systematically reviewing the literature. The topic is very important in clinical decision making. The authors performed a hard-working process. The manuscript follows the AMSTAR and PRISMA guidelines in most of the aspects. However, I also suggest major revision before considering publication. Besides the reviewers’ suggestions, I recommend that the manuscript needs to be edited regarding English language grammar and spelling. Secondly, the review seems to be conducted including only English-language papers and the review was run through 4 databases which are not as large as EMBASE database. The reviewers contributed in the selection process was not mentioned in the text, at least the capital letters of them should be given in parenthesis in methods section. These issues which are important limitations for a systematic review and possible influence of them to the results should be discussed in a separate limitation paragraph added at the end of discussion section.

Reviewer's Responses to Questions

**Comments to the Author**

1. Is the manuscript technically sound, and do the data support the conclusions?

Reviewer #1: Partly

Reviewer #2: Partly

2. Has the statistical analysis been performed appropriately and rigorously?

Reviewer #1: Yes

Reviewer #2: Yes

3. Have the authors made all data underlying the findings in their manuscript fully available?

Reviewer #1: No

Reviewer #2: Yes

4. Is the manuscript presented in an intelligible fashion and written in standard English?

Reviewer #1: No

Reviewer #2: No

Reviewer #1: Thank you very much for giving me the opportunity to comment on this important subject. The authors have made a great effort in screening the literature regarding the natural history of asymptomatic gallstones to assess the incidence of non-complicated and complicated gallstone disease and to identify possible risk factors. This is very important in clinical decision making.

The authors describe the strict rules as regard screening and selecting the relevant articles. Furthermore, they have used appropriate methods for summarizing the results. This is very impressive.

But I have several major concerns, which should be addressed before eventual publication:

1. The authors claim to have identified 9 studies fulfilling their inclusion criteria. Actually, it is only 8 studies as ref 26 and 29 (the Danish study) represent two articles of the same study – one is dealing with incidence and the other with risk factors.

2. I miss a short description of why the 108 studies were excluded. E.g. the MICOL study (Festi – ref 1 in the manuscript) is a huge screening study with ultrasonography of several Italian populations, where they found 580 persons with asymptomatic gallstones and followed them for 8.7 years. Why is this study not included? The Italians also conducted other studies on populations with ultrasonography to detect gallstones (GREPCO, Sirmona)

3. The study of Yousef Thwayeb from 2004 cannot be identified on NIH website (the reference in the manuscript is also insufficient), so I cannot comment on this study.

4. The studies included are extremely difficult to compare. Only the Swedish and the Danish studies are screening with ultrasonography of a random selected population and therefore should be the gold standard for assessing unselected persons with asymptomatic gallstones. The remaining study (here I don’t know about Thwayeb) are more or less clinical studies where people have attended a clinic/hospital maybe due to symptoms or worries. How sure can we be about the symptomatic state of these persons? E.g. ref 13 is just generated from a huge hospital material with no assessment of whether the gallstones were asymptomatic or not. No wonder that incidence of symptom development is very different in the varies studies.

5. Why were only three studies (ref 13, 28 and 30) included in the meta-analyses of incidence after 5, 10 and 15 years when seven studies (according to the table in the manuscript) provided incidence data for some of the time periods?

6. In the overview table of identified studies, the authors should mention both number examined and number with asymptomatic gallstones.

7. A major difference between the Danish study and the remaining is that the participants in the Danish study were blinded, meaning that the participants were not informed about their gallstone status. Given that most people develop abdominal symptoms now and then, the knowledge of having a gallstone may influence their decision on seeking a doctor, which more easily could turn an “asymptomatic” gallstone into a “symptomatic”.

8. It is very confusing to read the manuscript due to the many sensitivity calculations described in detail in the text. Could some of this text be transferred to an appendix making the main manuscript more readable?

Reviewer #2: The authors present an interesting topic.

However, below are some notes for improvement:

- Abstract section: Abstract is missing PRISMA-level standard elements, such as a clear statement of PICO components, type of effect estimates used for each major result, direction and magnitude of effects (reporting only p-values is insufficient for transparency, consider including RR values or at least the direction of effect).

- Introduction section: several epidemiological points are presented, but the flow is somewhat disjointed, and certain statements feel repetitive or unfocused. The section should better distinguish between background prevalence data, the clinical significance of ASG, and the rationale for conducting the review. Important concepts such as the natural history of ASG and the burden on healthcare systems are mentioned but not fully synthesized into a clear problem statement. Some epidemiological statistics are introduced without sufficient context or without explaining their relevance to the study objectives. The introduction would be strengthened by explicitly articulating the knowledge gap and clearly connecting that gap to the need for a systematic review. In addition, the paragraph is overly long and mixes aims with general commentary; it should be tightened to clearly and succinctly present the study’s objectives.

- Line 135: “A Single arm analysis, a proportion of complication was conducted using generalized linear mixed method” could be rephrased for clarity.

- Line 143: I² interpretation categories differ between guidelines. It stated <25% as low, 25–50% moderate, >75% high. This leaves a gap: 50–75% is not labeled.

- Section 3.1: Please consistent in terminology. The section uses “articles,” “studies,” and “records” interchangeably. PRISMA recommends consistent labeling (typically “records,” “reports,” “studies”).

- Section 3.2: Several sentences are grammatically incomplete or unclear.

- Discussion section: Several sentences are too narrative or conversational, which weakens scientific credibility (e.g. line 276-278, line 290, line 344). A discussion should remain neutral, objective, and concise.

- Conclusion section: It requires revision to improve clarity and coherence, as several sentences are long, fragmented, or grammatically incorrect. The overall structure lacks smooth transitions between key ideas such as risk factors, complications, and their clinical significance. Some statements, including those suggesting a “protective effect,” should be expressed more cautiously to avoid overstating the evidence. The conclusion does not clearly restate the main message of the review and not adequately acknowledge important limitations such as heterogeneity and the observational nature of the included studies.

I hope that these few remarks will help you to improve your manuscript.

Thank you.

**Do you want your identity to be public for this peer review?** For information about this choice, including consent withdrawal, please see our Privacy Policy

Reviewer #1: No

Reviewer #2: No

---

## [Author Response · Author response to Decision Letter 1]

31 Dec 2025

Dear Editorial Team,

We sincerely thank you and the reviewers for your feedback on our manuscript entitled “Asymptomatic Gallstones: Cumulative Incidence Proportion, Incidence Rate, and Risk Factors for Symptoms Development: Systematic Review and Meta-Analysis”, (Manuscript number: PONE-D-25-54208). We have carefully considered all your suggestions and revised the study accordingly. Below, we provide a point-by-point response to each comment.

Point by point responses:

Editor comments:

1. Style Requirements.

2. lines 162-165, remove.

3. English language grammar and spelling.

4. Secondly, the review seems to be conducted including only English-language papers and the review was run through 4 databases which are not as large as EMBASE database.

5. The reviewers contributed in the selection process.

6. These issues which are important limitations for a systematic review and possible influence of them to the results should be discussed in a separate limitation paragraph added at the end of discussion section.

Answers to editor’s comments:

1. Ans: style requirements have been revised.

2. Ans: lines 162-165 have been removed.

3. Ans: The manuscript has been revised regarding English language grammar and spelling.

4. Ans: Thank you for raising this point, our rationale for using these data bases was because SCOPUS is considered the largest publication library and we searched through its database along with PubMed, Web Of Science, and Science Direct which all together create a much larger collective library. In addition, according to the systematic review frameworks suggested by PRISMA guidelines, a comprehensive search does not require using all databases available, rather it requires adequate coverage of the relevant literature, which we achieved by manual searching through the references of all the studies that were subjected to full text screening, by this we created an effect similar to snow-balling technique for collecting all relevant studies in the literature. Also, we do not have an access to EMBASE.

5. Ans: The initials of each author that contributed to the screening process have been added.

6. Ans: limitations have been modified and added to the ending of the discussion part.

Reviewer #1:

1. “The authors claim to have identified 9 studies fulfilling their inclusion criteria. Actually, it is only 8 studies as ref 26 and 29 (the Danish study) represent two articles of the same study – one is dealing with incidence and the other with risk factors.”

Response: We thank the reviewer for this careful observation. We are aware that references 26 and 29 come from the same study, but as their outcomes are different and each was analyzed separately, we treated them as separate studies. Technically, we included 8 studies reported in 9, two of which originate from the same study. This point has been clarified in the manuscript, and the results and abstract sections along with the PRISMA flow chart have been updated accordingly to reflect this important distinction.

2. “I miss a short description of why the 108 studies were excluded. E.g. the MICOL study (Festi – ref 1 in the manuscript) is a huge screening study with ultrasonography of several Italian populations, where they found 580 persons with asymptomatic gallstones and followed them for 8.7 years. Why is this study not included? The Italians also conducted other studies on populations with ultrasonography to detect gallstones (GREPCO, Sirmona)”

Response: We thank the reviewer for this observation. The primary aim of our meta-analysis was to pool cumulative incidence of symptom development and to evaluate risk factors distinguishing patients who remained asymptomatic from those who became symptomatic during follow-up. Therefore, only observational studies that reported cumulative incidence data and/or comparative risk-factor analyses were eligible.

Large population-based screening cohorts such as the MICOL study (Festi et al.), GREPCO, and Sirmione primarily provided prevalence estimates and descriptive follow-up data but did not report cumulative incidence in a form compatible with our predefined pooling approach, nor did they present comparative risk-factor outcomes between asymptomatic and symptomatic groups. For these reasons, these studies were excluded at full-text review.

3. “The study of Yousef Thwayeb from 2004 cannot be identified on NIH website (the reference in the manuscript is also insufficient), so I cannot comment on this study.”

Response: The PDF of the full-text study is attached with the revised submission, to be reviewed.

4. “The studies included are extremely difficult to compare. Only the Swedish and the Danish studies are screening with ultrasonography of a random selected population and therefore should be the gold standard for assessing unselected persons with asymptomatic gallstones. The remaining study (here I don’t know about Thwayeb) are more or less clinical studies where people have attended a clinic/hospital maybe due to symptoms or worries. How sure can we be about the symptomatic state of these persons? E.g. ref 13 is just generated from a huge hospital material with no assessment of whether the gallstones were asymptomatic or not. No wonder that incidence of symptom development is very different in the varies studies.”

Response: We agree that the articles included have different study designs as well as differences in the population selection techniques, which does indeed affect comparability. Swedish, Danish, Spanish (Thawayb) are population-based screening studies using the most robust assessment of truly asymptomatic gallstones carriers, as they used ultrasonography in randomly selected individuals regardless of symptoms. In contrast, the rest of articles where hospital based studies, this issue was addressed by careful selection of the non-standardized studies. After careful revision, we got reassured after double checking the Moris 2023 study (ref 13), which specifically mentioned the following in the methods section: (The NLP searched for regular expressions or words, phrases, and combinations that are used in the imaging notes to indicate stones or that patients are asymptomatic). Moreover, in the discussion section during data interpretation: (In this observational cohort study of more than 22,000 patients with asymptomatic gallstones incidentally discovered on imaging, 10% of patients developed symptoms within 5 years and 22% within 10 years), the study even referred to asymptomatic gallstones as AGs, and symptomatic ones as SGs. This and after revising the other similar studies to Moris 2023, we are assured that no ambiguous study was included in the analysis. To overcome the problem of differences of study population, subgrouping analysis is solution. However, this method could not be conducted in our study because each outcome would only be mentioned in at most 3 studies, and the results reporting of the existing studies was not helpful in finding sufficient comparable factors between the studies.

5. “Why were only three studies (ref 13, 28 and 30) included in the meta-analyses of incidence after 5, 10 and 15 years when seven studies (according to the table in the manuscript) provided incidence data for some of the time periods?”

Response: The cumulative incidence proportion was pooled using a Generic Inverse Variance meta-analysis. Only three studies (Halldestam I, 2004; Morris-Stiff, 2023; Gracie, 1982) could be included because they reported the effect size along with the corresponding standard error (SE). The other studies provided only the effect size, without SE, and are still summarized in Table 1 for consistency. The discussion section was modified accordingly to explain this point.

6. “In the overview table of identified studies, the authors should mention both number examined and number with asymptomatic gallstones.”

Response: Those numbers are addressed as total cohort size representing the number of examined population, and sample size of asymptomatic representing the asymptomatic gallstones population. In addition to the sample size of symptomatic gallstones patients at baseline.

7. “A major difference between the Danish study and the remaining is that the participants in the Danish study were blinded, meaning that the participants were not informed about their gallstone status. Given that most people develop abdominal symptoms now and then, the knowledge of having a gallstone may influence their decision on seeking a doctor, which more easily could turn an “asymptomatic” gallstone into a “symptomatic”.”

Response: We performed sensitivity analysis excluding the Danish study to evaluate its influence. The Danish study contributed data to the following outcomes: (1) risk factors (sex, smoking, and age), (2) complications (biliary pain, common bile duct stones, acute cholecystitis, and gallstone pancreatitis), and (3) incidence rate.

After exclusion, the associations for male sex (protective), female sex (not significant), and age (not significant) remained consistent with the primary analysis. In contrast, smoking became a statistically significant risk factor for symptom progression following exclusion. The pooled incidence rate remained unchanged and aligned with the primary results.

Regarding complications, removal of the Danish study resulted in higher pooled proportions for several outcomes, including biliary pain (0.75), acute cholecystitis (0.21), and gallstone pancreatitis (0.12), indicating that inclusion of the Danish study led to more conservative estimates for these complications.

8. “It is very confusing to read the manuscript due to the many sensitivity calculations described in detail in the text. Could some of this text be transferred to an appendix making the main manuscript more readable?”

Response: Thank you for this suggestion. We moved the majority of the sensitivity analysis and publication-bias descriptions to the Supplementary File under the sections “Leave-one-out Sensitivity Analysis Results” and “Publication Bias Results”.

Reviewer #2:

1. “Abstract section: Abstract is missing PRISMA-level standard elements, such as a clear statement of PICO components, type of effect estimates used for each major result, direction and magnitude of effects (reporting only p-values is insufficient for transparency, consider including RR values or at least the direction of effect).”

Response: Thank you for this important point. To improve transparency, we have revised the abstract by: (1) clearly stating the objective in PICO format, and (2) reporting the effect size for each major outcome, including the corresponding 95% confidence intervals and p-values. These modifications ensure that the abstract conveys not only statistical significance but also the direction and magnitude of effects, consistent with PRISMA reporting standards.

2. “Introduction section: several epidemiological points are presented, but the flow is somewhat disjointed, and certain statements feel repetitive or unfocused. The section should better distinguish between background prevalence data, the clinical significance of ASG, and the rationale for conducting the review. Important concepts such as the natural history of ASG and the burden on healthcare systems are mentioned but not fully synthesized into a clear problem statement. Some epidemiological statistics are introduced without sufficient context or without explaining their relevance to the study objectives. The introduction would be strengthened by explicitly articulating the knowledge gap and clearly connecting that gap to the need for a systematic review. In addition, the paragraph is overly long and mixes aims with general commentary; it should be tightened to clearly and succinctly present the study’s objectives.”

Response: We thank the reviewer for such an important notice, which we as a consequence took all into consideration and repeated the majority of the introduction section in a flow consistent with these suggestions and addressing all of the raised issued regarding this topic.

3. “Line 135: “A Single arm analysis, a proportion of complication was conducted using generalized linear mixed method” could be rephrased for clarity.”

Response: Thank you for this suggestion. We have rephrased the sentence for clarity as follows: “A single-arm meta-analysis was performed to estimate complication proportions and the cumulative incidence proportion for progression to symptomatic gallstone disease, using a generalized linear mixed-effects model.”

4. “Line 143: I² interpretation categories differ between guidelines. It stated <25% as low, 25–50% moderate, >75% high. This leaves a gap: 50–75% is not labeled.”

Response: Thank you for this comment. We have updated the I² interpretation in the manuscript to align with the latest guidance from the Cochrane Handbook, which advises using the following approximate ranges: 0–40% (might not be important), 30–60% (moderate heterogeneity), 50–90% (substantial heterogeneity), and 75–100% (considerable heterogeneity). This revision removes the previous gap and reflects the most recent Cochrane recommendations.

5. “Section 3.1: Please consistent in terminology. The section uses “articles,” “studies,” and “records” interchangeably. PRISMA recommends consistent labeling (typically “records,” “reports,” “studies”).”

Response: Thank you for this notice. It has been taken into consideration and addressed.

6. “ Section 3.2: Several sentences are grammatically incomplete or unclear.”

Response: Thank you, corrected.

7. “Discussion section: Several sentences are too narrative or conversational, which weakens scientific credibility (e.g. line 276-278, line 290, line 344). A discussion should remain neutral, objective, and concise.”

Response: Thank you, it has been edited accordingly.

8. “Conclusion section: It requires revision to improve clarity and coherence, as several sentences are long, fragmented, or grammatically incorrect. The overall structure lacks smooth transitions between key ideas such as risk factors, complications, and their clinical significance. Some statements, including those suggesting a “protective effect,” should be expressed more cautiously to avoid overstating the evidence. The conclusion does not clearly restate the main message of the review and not adequately acknowledge important limitations such as heterogeneity and the observational nature of the included studies.”

Response: Thank you, it has been edited.

---

## [Decision Letter · Decision Letter 1]

1 Feb 2026

Asymptomatic Gallstones: Cumulative Incidence Proportion, Incidence Rate, and Risk Factors for Symptoms Development: Systematic Review and Meta-Analysis

PLOS One

Dear Dr. Shatnawi,

Thank you for submitting your manuscript to PLOS ONE. After careful consideration, we feel that it has merit but does not fully meet PLOS ONE’s publication criteria as it currently stands. Therefore, we invite you to submit a revised version of the manuscript that addresses the points raised during the review process.

**ACADEMIC EDITOR:** The revised manuscript follows the AMSTAR and PRISMA guidelines. I believe that this study merits consideration for publication in this journal, as it represents an impressive and comprehensive work. However, the revisions suggested by the reviewer must be addressed beforehand. Specifically, the inclusion and exclusion criteria should be meticulously re-evaluated, and the potential limitations arising from these criteria must be discussed. It is crucial to emphasize that factors ranging from the geographical distribution of the included studies to the specific diagnostic modalities employed may significantly impact the interpretation of the analysis. Addressing these points within the Discussion section will substantially enhance the overall quality and clinical relevance of the manuscript.

We look forward to receiving your revised manuscript.

Kind regards,

Ozlem Boybeyi

Academic Editor

PLOS One

**Journal Requirements:**

**Additional Editor Comments:**

I believe that this study merits consideration for publication in this journal, as it represents an impressive and comprehensive work. However, the revisions suggested by the reviewer must be addressed beforehand. Specifically, the inclusion and exclusion criteria should be meticulously re-evaluated, and the potential limitations arising from these criteria must be discussed. It is crucial to emphasize that factors ranging from the geographical distribution of the included studies to the specific diagnostic modalities employed may significantly impact the interpretation of the analysis. Addressing these points within the Discussion section will substantially enhance the overall quality and clinical relevance of the manuscript.

Reviewers' comments:

Reviewer's Responses to Questions

**Comments to the Author**

Reviewer #1: All comments have been addressed

Reviewer #2: All comments have been addressed

2. Is the manuscript technically sound, and do the data support the conclusions?

Reviewer #1: Partly

Reviewer #2: Yes

3. Has the statistical analysis been performed appropriately and rigorously?

Reviewer #1: Yes

Reviewer #2: Yes

4. Have the authors made all data underlying the findings in their manuscript fully available?

Reviewer #1: No

Reviewer #2: Yes

5. Is the manuscript presented in an intelligible fashion and written in standard English?

Reviewer #1: Yes

Reviewer #2: Yes

Reviewer #1: Thank you for giving me the opportunity to review the revised manuscript.

First I would like to acknowledge the huge work the authors have performed. It is very important.

My main problem is still the selection of the articles for the review - or more precisely the assessment of these articles. As I mentioned in my first review the studies are very different ranging from screening of general populations to surgical case-series. Whereas it is quite straight forward to identify asymptomatic gallstones in screened population I still have my doubt about the clinical series. They maybe can be used as support for findings in the population based studies, but not regarded equal to them. In their answer the authors have a long explanation why they find the clinical series to fullfill the criteria for dealing with asymptomatic gallstones, but these arguments should be described in the manuscript. It is of course elegant to use "leave-one-out" approach, but this does not help as regard the quality of the various studies.

Thank you for provinding me with the Spanish study, but please fill in the correct reference in the Reference-section.

I think I pointed out that it seems as if the Danish study was blinded. I find no comments on this in the revised manuscript, which I find a pity, as this makes the Danish study very important as regard the natural history of asymptomatic gallstones. Given the knowledge of having gallstones could bias people in the interpretation of the reason for abdominal symptoms and thereby change the status of asymptomatic to symptomatic gallstones. As you know there is no clearcut definition of gallstone symptoms.

The text has improved after removal of part of the results to the suplementary section.

Minor things:

Use of oral contraceptive pills is mentioned as an exclusion criteria. Many women used AC pills during these study periods and maybe it is not mentioned in the manuscript. Suggest you leave out this criteria

Incidence rate 0.02 - I miss a timeframe.

Why do the authers both write 95 % conf. int. and a p-value? conf. int. should be sufficient

Male gender is protective, but female gender is not a risk factor? Difficult to understand. Please explain.

Reviewer #2: Thank you for your hardwork and thorough revision. I appreciate the way you addressed all of the previous comments. The manuscript has improved significantly in clarity, structure, and completeness. I have no further major concerns and believe the paper is now suitable for publication pending the journal’s final editorial checks.

**Do you want your identity to be public for this peer review?** For information about this choice, including consent withdrawal, please see our Privacy Policy

Reviewer #1: No

Reviewer #2: No

---

## [Author Response · Author response to Decision Letter 2]

2 Mar 2026

Dear Editorial Team,

We sincerely thank you and the reviewers for your feedback on our manuscript entitled “Asymptomatic Gallstones: Cumulative Incidence Proportion, Incidence Rate, and Risk Factors for Symptoms Development: Systematic Review and Meta-Analysis”, (Manuscript number: PONE-D-25-54208). We have carefully considered all your suggestions and revised the study accordingly. Below, we provide a point-by-point response to each comment.

Point by point responses:

ACADEMIC EDITOR:

• The inclusion and exclusion criteria should be meticulously re-evaluated, and the potential limitations arising from these criteria must be discussed. It is crucial to emphasize that factors ranging from the geographical distribution of the included studies to the specific diagnostic modalities employed may significantly impact the interpretation of the analysis. Addressing these points within the Discussion section will substantially enhance the overall quality and clinical relevance of the manuscript.

- Author’s response: We thank the academic editor for the insightful points raised, we re-evaluated the inclusion and exclusion criteria per the reviews comments, and refined it accordingly.

- We thank the editor again for the insightful comments regarding the imapct of the geographical variation and diagnostic methods, we furtherly elaborated on these issues and the limitaions araising from each in the discussion section.

Reviewer’s Comment

Reviewer #1:

• Have the authors made all data underlying the findings in their manuscript fully available?

Reviewer # 1: No

- Author’s Response: During the submission process, we declared that all data used to generate this paper will be available per request, and we are happy to attached our data extraction sheet to the supporting files for the reviewer’s perusal to insure transparency.

• My main problem is still the selection of the articles for the review - or more precisely the assessment of these articles. As I mentioned in my first review the studies are very different ranging from screening of general populations to surgical case-series. Whereas it is quite straight forward to identify asymptomatic gallstones in screened population I still have my doubt about the clinical series. They maybe can be used as support for findings in the population based studies, but not regarded equal to them. In their answer the authors have a long explanation why they find the clinical series to fullfill the criteria for dealing with asymptomatic gallstones, but these arguments should be described in the manuscript. It is of course elegant to use "leave-one-out" approach, but this does not help as regard the quality of the various studies.

- Author’s Response: We thank the reviewer for this insightful comment, we added the argument regarding considering these studies sufficient in the discussion section and discussed the limitations arising from the differences between studies ((It is evident that the articles included in this study have different study designs ….. that no ambiguous study was involved in the analysis)), and ((Heterogeneity is also explained by timely influenced evolution …… research nationwide databases)).

• Thank you for provinding me with the Spanish study, but please fill in the correct reference in the Reference-section.

- Author’s Response: It has been added, reference no. 32.

• I think I pointed out that it seems as if the Danish study was blinded. I find no comments on this in the revised manuscript, which I find a pity, as this makes the Danish study very important as regard the natural history of asymptomatic gallstones. Given the knowledge of having gallstones could bias people in the interpretation of the reason for abdominal symptoms and thereby change the status of asymptomatic to symptomatic gallstones. As you know there is no clearcut definition of gallstone symptoms.

- Author’s Response: Thank you for drawing attention to this point, indeed it is inevitable that the danish study is of higher quality when compared to the other studies, we have included a part in the discussion explaining why the analysis concluded that the danish study was a source of bias even though it is the highest quality paper, and how is that carefully interpreted in our study, ((However, the danish study Shabanzadeh et al.2017 ……strongly emphasizes the need for further studies)).

• Use of oral contraceptive pills is mentioned as an exclusion criteria. Many women used AC pills during these study periods and maybe it is not mentioned in the manuscript. Suggest you leave out this criteria

- Author’s Response: We appreciate the reviewer’s insightful comment. We agree that the use of oral contraceptive pills may have been underreported in the included studies and could have introduced unnecessary exclusion. Therefore, this criterion has been removed from the exclusion criteria in the revised manuscript. The eligibility section has been updated accordingly.

• Incidence rate 0.02 - I miss a timeframe.

- Author’s Response: Thank you for this observation. We have revised the manuscript to specify that the incidence rate is reported per person-year in the main text to ensure clarity and appropriate interpretation.

• Why do the authers both write 95 % conf. int. and a p-value? conf. int. should be sufficient

- Author’s Response: Thank you for this comment. We agree that reporting 95% confidence intervals is sufficient to convey the precision and statistical significance of the results. Therefore, p-values have been removed from the text in the revised manuscript.

• Male gender is protective, but female gender is not a risk factor? Difficult to understand. Please explain.

- Author’s Response: Thank you for raising this point, as it is indeed a very critical point that needs to be fully explained to avoid confusion, we added detailed explanations for this issue in the discussion section, (( However, it is important to emphasize on that this study is investigating the development of symptoms in an individual with existent asymptomatic gallstones……by affecting the etiological mechanisms behind gallstones)), and ((there is no scientific evidence proven on why male gender may be a protective factor……. relative to the male population percentages that were reported in the other included studies)).

Reviewer #2:

• Thank you for your hardwork and thorough revision. I appreciate the way you addressed all of the previous comments. The manuscript has improved significantly in clarity, structure, and completeness. I have no further major concerns and believe the paper is now suitable for publication pending the journal’s final editorial checks.

- Author’s Response: We would like to thank the reviewer for their input into improving the quality and comprehensiveness of our study.

---

## [Editor Report · Decision Letter 2]

5 Mar 2026

Asymptomatic Gallstones: Cumulative Incidence Proportion, Incidence Rate, and Risk Factors for Symptoms Development: Systematic Review and Meta-Analysis

PONE-D-25-54208R2

Dear Dr. Shatnawi,

We’re pleased to inform you that your manuscript has been judged scientifically suitable for publication and will be formally accepted for publication once it meets all outstanding technical requirements.

Kind regards,

Ozlem Boybeyi

Academic Editor

PLOS One

Additional Editor Comments (optional):

The revised manuscript fulfills all the requirements, and the authors addressed all of the comments. I think it is an impressive and comprehensive work, and I suggest publication in the Journal.
---

## [Editor Report · Acceptance letter]

PONE-D-25-54208R2

PLOS One

Dear Dr. Shatnawi,

I'm pleased to inform you that your manuscript has been deemed suitable for publication in PLOS One. Congratulations! Your manuscript is now being handed over to our production team.

Kind regards,

on behalf of

Professor Ozlem Boybeyi

Academic Editor

PLOS One